# Towards Predicting Length of Stay and Identification of Cohort Risk Factors Using Self-Attention-Based Transformers and Association Mining: COVID-19 as a Phenotype

**DOI:** 10.3390/diagnostics13101760

**Published:** 2023-05-17

**Authors:** Fakhare Alam, Obieda Ananbeh, Khalid Mahmood Malik, Abdulrahman Al Odayani, Ibrahim Bin Hussain, Naoufel Kaabia, Amal Al Aidaroos, Abdul Khader Jilani Saudagar

**Affiliations:** 1Department of Computer Science & Engineering, Oakland University, 115 Library Drive, Rochester, MI 48309, USA; fakharealam@oakland.edu (F.A.); oananbeh@oakland.edu (O.A.); 2Infection Control Center of Excellence Prince Sultan Military Medical City, Riyadh 12233, Saudi Arabia; aalodayani@psmmc.med.sa (A.A.O.); ihussain@kfshrc.edu.sa (I.B.H.); nkaabia@psmmc.med.sa (N.K.); amalalaidaroos@gmail.com (A.A.A.); 3Information Systems Department, College of Computer and Information Sciences, Imam Mohammad Ibn Saud Islamic University (IMSIU), Riyadh 11432, Saudi Arabia; aksaudagar@imamu.edu.sa

**Keywords:** deep learning, COVID-19, clinical informatics, machine learning, transformer, association mining

## Abstract

Predicting length of stay (LoS) and understanding its underlying factors is essential to minimizing the risk of hospital-acquired conditions, improving financial, operational, and clinical outcomes, and better managing future pandemics. The purpose of this study was to forecast patients’ LoS using a deep learning model and to analyze cohorts of risk factors reducing or prolonging LoS. We employed various preprocessing techniques, SMOTE-N to balance data, and a TabTransformer model to forecast LoS. Finally, the Apriori algorithm was applied to analyze cohorts of risk factors influencing hospital LoS. The TabTransformer outperformed the base machine learning models in terms of F1 score (0.92), precision (0.83), recall (0.93), and accuracy (0.73) for the discharged dataset and F1 score (0.84), precision (0.75), recall (0.98), and accuracy (0.77) for the deceased dataset. The association mining algorithm was able to identify significant risk factors/indicators belonging to laboratory, X-ray, and clinical data, such as elevated LDH and D-dimer levels, lymphocyte count, and comorbidities such as hypertension and diabetes. It also reveals what treatments have reduced the symptoms of COVID-19 patients, leading to a reduction in LoS, particularly when no vaccines or medication, such as Paxlovid, were available.

## 1. Introduction

Health system infrastructure worldwide was severely strained by the rapid surge of patients infected with different coronavirus variants, and many countries struggled to provide basic healthcare and timely services to patients [1]. Despite the availability of COVID-19 vaccines, statistics show that the hospitalization rate spiked globally in the winter seasons during the last two years [2]. According to a study conducted in the U.S., inadequate critical care is associated with resource availability [3], and an hour delay in services is associated with a 3% increase in patient mortality [4]. An extended LoS is associated with a high risk of negative outcomes including adverse drug effects, hospital-acquired infections, inadequate nutritional levels, and many other complications [5]. Inpatient care accounts for roughly a third of all healthcare spending in the U.S., with an average length of stay of 4.5 days and a daily cost of USD 10,400 [6]. Predicting length of stay (LoS) and understanding how to reduce it are the most critical factors for optimal usage of hospital infrastructure and medical resources during the emergence of new infectious diseases, improving financial, operational, and clinical outcomes by reducing costs for patients, such as facility expenses, supplies, and staffing. In addition, this minimizes the risk of hospital-acquired infections and reduces the wait time for patients. Thus, for precise resource management and utilization of the current infrastructure of the healthcare, a sophisticated approach is needed to predict the LoS, identify the cohort risk factors that lead to increased LoS, and determine what treatments can reduce the LoS.

LoS prediction was performed for different diseases using statistical and conventional machine learning (ML) techniques such as logistic regression (LR), a random forest classifier (RFC), decision trees (DTs), etc. For example, Luo et al. [7] used LR and an RFC to predict LoS in patients with pulmonary disease. Likewise, Dogu et al. [8] employed artificial neural networks to predict LoS in chronic obstructive pulmonary disease (COPD) patients while Kulkarni et al. [9] performed this using a multilayer perceptron (MLP) for acute coronary syndrome patients. Likewise, to predict ICU admission, mortality, and survivors’ LoS for COVID-19, Dan et al. [10] created three ML prediction models without identifying cohorts of risk factors, and analysis was based only on univariate analysis. Lastly, Vekaria et al. [11] developed statistical techniques such as truncation correction and survival bias to predict COVID-19 patients’ LoS, but the employed data are not multimodal and suffer from quality issues, such as a missing timeline for discharged patients, and a limitation of the statistical model in recognizing hidden patterns.

Deep learning (DL) has proven capable of extracting complex, hidden correlations from data and has achieved promising results when compared to existing ML methods. For instance, Zebin and Chaussalet [12] used an autoencoder deep neural network to categorize short stays (0–7 days) and long stays (>7 days) using the Medical Information Mart for Intensive Care III (MIMIC III) dataset [13], but the dataset lacked multimodalities and the methodology does not involve analysis of the cohort of risk factors responsible for extended LoS. Likewise, Harerimana et al. [14] proposed a self-attention-based DL method to predict LoS and in-hospital mortality, but this method also has the limitation of using limited lab data and suffers from the lack of availability of radiological information. Rajkomar et al. [15] proposed a three-tier approach by combining three DL models to predict hospital readmission and patients’ LoS based on data belonging to patients with varying diseases, without identifying the cohort of risk factors affecting patient LoS. These existing approaches only classify discharge or admit cases and predict the duration of the LoS using single-modality data and are unable to provide the cohort of influencing factors that could increase or decrease patients’ LoS, and most of them were not designed for predicting LoS for patients with infectious diseases.

The study design in this research focuses on a detailed analysis of interaction and association between different risk factors and develops a framework for predicting the LoS of COVID-19 patients using a state-of-the-art transformer-based architecture and identifies groups of influencing factors occurring together that affect LoS using pattern recognition techniques by mining multimodal patient data such as the lab data and X-ray data of COVID-19 patients.

In this paper, we propose a state-of-the-art self-attention-based TabTransformer model that utilizes multiple modalities, including clinical features, patient demographic data, and X-ray reports, to accurately predict patient length of stay (LoS). Secondly, we present a framework within which the result of machine-learning-based methods for LoS prediction can be analyzed with association mining rules to identify the cohort of risk factors affecting the LoS in hospitals.

The remainder of this paper is organized as follows: Section 2 describes the proposed method while Section 3 presents experimental results and evaluation. Section 4 presents further discussion on the obtained results. Section 5 discusses limitations and future research directions, followed by the conclusion in Section 6.

## 2. Materials and Methods

The proposed framework for LoS prediction and the identification of cohorts of risk factors includes three major components: data acquisition, data preparation, and LoS risk modeling. The detailed architecture and various subcomponents are shown in Figure 1. The main tasks in the data acquisition module include obtaining de-identified patient data from electronic health records (EHRs), and identification of missing risk factors in the initial patients’ data with respect to main prognostic factors. The data preparation module includes data cleaning, quantization, and balancing the data with respect to output (deceased, discharge) and LoS to enable an equal ratio in each category. The LoS risk modeling includes building and hypertuning machine learning models and using association mining algorithms to identify cohorts of risk factors.

### 2.1. Data Acquisition

This study was approved by the Institutional Review Board (IRB) at Imam Mohammad Ibn Saud Islamic University. After approval (No. 42-2021), a total of 311 de-identified patient records from an in-house registry of patients admitted to the ICU from Prince Sultan Military Medical City Infection Control Center of Excellence from April 2020 to January 2021 were queried retrospectively. After analyzing the data using prognostic factors, admission, and discharged date, 308 cases were included in the analysis. Since the data were de-identified and retrospective, and the study conducted was not an intervention study, patient consent was not required. This dataset includes 60 patients who died during treatment and 248 patients who were discharged.

To ensure the quality of the data, we followed a three-step process. At first, we curated the data from medical electronic files using a study design prepared by infectious disease experts. Secondly, it was manually verified by the trained physicians, and at last, the final dataset was validated and approved by the infectious disease specialist. In total, 89 features from three different categories—general information, X-ray, and lab tests—were extracted. Table 1 shows the descriptions of available features and frequencies with respect to different modalities of data.

### 2.2. Data Preparation

In the data preparation stage, we performed basic data preprocessing such as binning, encoding, and initial exploratory data analysis (EDA) by examining the distribution of attributes and summarizing the statistics of the data to uncover hidden patterns. Table 2 shows the descriptive statistics for prominent features across different modalities of data and the proportion of patients with comorbidities such as hypertension, diabetes, etc., or specific symptoms such as fever and shortness of breath. Additionally, during the EDA, we found that certain tests, such as those measuring levels of aspartate aminotransferase, creatine, phosphokinase, and fibrinogen, were not requested for some of the patients. Therefore, these features were not considered in modeling.

#### 2.2.1. Natural Binning

Data binning is a method used to minimize the effect of small observation errors. This method is used to discretize continuous features and transform them into categorical features. We analyzed the distribution of all the continuous features, such as age, pH, PaO_2_, HCO_3_, etc., and performed natural binning using Jenks–Caspell [16] natural breaks, and then consulted with medical experts to adjust the binning boundaries. Binning introduces non-linearity and improves the performance of machine learning models by minimizing small observation errors. Table A1 in Appendix A shows the optimized categorization for the continuous features.

#### 2.2.2. Encoding of Categorical Features

Our dataset comprises various categorical features, such as diabetes and hypertension, including label values of (0-No, 1-Yes). It is essential that categorical features be transformed into numeric features before ML models can be trained effectively. We utilized an internal one-hot encoder and converted various categorical features to numerical features so that ML models can process them efficiently.

#### 2.2.3. LoS Category Creation

Using the combined dataset from the data acquisition step, we calculated the LoS using the admission and discharge or mortality timelines for all the patients and divided it into categories such as deceased within 3 weeks, deceased after 3 weeks, discharged within 1 week, discharged between 1 and 2 weeks, discharged between 2 and 3 weeks, discharged between 3 and 4 weeks, and finally, discharged after 4 weeks from the date of admission. Table 3 shows the number of patients in each of these LoS categories and patient frequency.

#### 2.2.4. Data Balancing with Respect to LoS

The original dataset was imbalanced considering the various LoS categories within the discharged and deceased datasets. A balanced dataset is necessary to train the machine learning model to generate higher-accuracy models and make unbiased decisions. The two primary approaches to making a balanced dataset out of an imbalanced dataset are undersampling and oversampling. Given the limited number of patients in each categorized LoS, in this work, we employed random oversampling using a variant of the synthetic minority oversampling technique (SMOTE) called SMOTE-N [17]. This technique works well for categorical data such as diabetes, hypertension, interstitial lung disease, bronchial asthma, liver disease, HIV, cirrhosis, and cardiomyopathies, and generates new instances from existing minority classes by taking samples of feature space for each target class and its nearest neighbors. This algorithm then generates new examples that combine features of the target case with features of its neighbors and increases the number of features available to each class, making the data more general. After balancing the data, the LoS categories within the discharged (n = 84) and deceased (n = 36) datasets contained an equal number of records. Table 4 shows the original data and the increased count of instances within each category after applying SMOTE-N.

### 2.3. LoS Risk Modeling

The preprocessed and balanced data were used to develop and train an LoS predictor model for both deceased and discharged patients. We developed an LoS predictor model (t-LoSP) using a state-of-the-art transformer-based classifier, followed by a cohort risk factor identifier (CRFI), to identify groups of risk factors affecting the patient LoS.

#### 2.3.1. t-LoS Predictor

The TabTransformer is an innovative and recently developed deep tabular data model that can be used for both supervised and semi-supervised learning. Self-attention transformers form the foundation of the TabTransformer model. In the dataset, we have many categorical variables, such as diabetes, hypertension, abnormal X-rays, etc. Other available machine learning models, such as neural networks, do not consider the interaction and relationships between categorical variables in the categorical embedding process. In the transformer-based architecture, the transformer layers convert categorical feature embeddings into strong contextual embeddings to improve prediction accuracy. The TabTransformer architecture consists of a column embedding layer, a stack of N transformer layers, and a multilayer perceptron. An individual transformer layer consists of a multi-head self-attention layer followed by a position-wise feed-forward layer. Figure 2 shows the detailed architecture of the self-attention-based TabTransformer model and Table A2 in Appendix A shows the hypertuned parameter values. The following are the steps for model execution:

Let x denote the input feature set and y the multiclass target variable. Feature set x consists of both categorical ($X_{ca}=\{x_{1},x_{2},x_{3}\ldots x_{m}\}$) and continuous variables ($X_{co}$). All categorical features are embedded into the embedding space of dimension d using column embedding.

Let $e\varphi_{ⅉ}\in R^{d}\;for\;j\in \{1,\cdot \cdot \cdot ,m\}$ be the embedding of the $x_{j}$ feature, and $E\varphi (X_{ca})=\{e\varphi_{1}(x_{1}),\cdot \cdot \cdot ,e\varphi_{m}\left(x_{m}\right)\}$ be the embeddings for all categorical features.

The set of projected categorical embeddings, $E\varphi \left(X_{ca}\right),$ are input to the first transformer layer, as shown in Figure 2. The output of the first transformer layer is sent to the next transformer layer and continues for N transformer layers. The embedding output from individual layers is transformed into contextual embedding when resulting from the top transformer layer via consecutive aggregation of context from other embeddings. The sequence of transformer layers is denoted as a function, $f_{\sigma }$. This function operates on parametric embeddings of categorical variables $\left\{e\varphi_{1}(x_{1}),\cdot \cdot \cdot ,e\varphi_{m}\left(x_{m}\right)\right\}$ and results in corresponding contextual embeddings $\left\{k_{1},k_{2}\cdot \cdot \cdot ,k_{m}\right\},where\;k_{i}\in R\;for\;i\in \{1,\cdot \cdot \cdot ,m\}$. In the end, the contextual embeddings obtained from transformer encoders $\left\{k_{1},k_{2}\cdot \cdot \cdot ,k_{m}\right\}$ are concatenated with the continuous features $X_{co}$ to form a vector of dimension (d×m+c) and serve as input for the MLP classifier, denoted by $h_{\psi }$, to compute the target prediction variable y. Let J be the categorical cross-entropy for the multiclass classification prediction task. We minimize the loss, $J\left(x,y\right),$ to learn all the parameters of the TabTransformer using the gradient descent optimization method. The parameters of the TabTransformer include σ for the transformer layers, φ for column embeddings, and ψ for the top MLP classifier.
(1)$$J\left(x,y\right)\equiv H\left(g\psi \left(f_{\sigma }\left(E\varphi \left(x_{ca}\right)\right),x_{co}\right),y\right)$$

More information about the TabTransformer architecture is available in [18]. The effectiveness of the TabTransformer for multiclass datasets, and particularly LoS prediction, is unknown. In each deceased and discharged dataset, 70% of the data are used to train the model, and 30% to test its accuracy across multiclass prediction of LoS.

#### 2.3.2. Cohort Risk Factor Identifier (CRFI)

The aim of the CRFI is to identify the factor or combination of risk factors that have the greatest influence on patient LoS in the hospital. For this purpose, we employed Apriori [19], which generates association rules by mining transactional data, which in our case include patient characteristics, symptoms, lab data, and X-ray features in each defined category of LoS. Association mining consists of the following four steps:

**STEP 1:** Find all frequent item sets, i.e., all patient characteristics appearing frequently together in the data with 50% support and 70% confidence.
(2)$${Support}_{\left(A\right)}=\frac{Number\;of\;transactions\;in\;which\;A\;appears}{Total\;number\;of\;transactions}$$
(3)$${Support}_{\left(A\to B\right)}=\frac{Number\;of\;transactions\;in\;which\;A\;and\;B\;appear\;together}{Total\;number\;of\;transactions}$$
(4)$${Confidence}_{A\to B}=\frac{Support(A\to B)}{Support\;(A)}$$
where A and B are item sets such as hypertension, diabetes, age ranges, lab characteristics, ranges, etc., as defined in Table 3.

**STEP 2:** Generate association rules from the aforesaid frequent itemset.

**STEP 3:** Create a metric by calculating the normalized harmonic mean of support and confidence using a min–max scalar.

**STEP 4:** Productionize the rule by selecting all the rules above the threshold value (β = 0.7). This threshold value is decided after the rules are reviewed by experts and considering the frequency of patients belonging to each rule.

## 3. Results and Evaluation

This section details the performance of the proposed framework and highlights the important features in estimating the LoS of hospital patients. The experiments were performed on a 2.10 GHz Intel(R) Xeon(R) Platinum 8160 processor in a Python programming environment.

### 3.1. COVID-19 Risk Model Results

We performed experiments with five ML models: AdaBoost (AB), a decision tree (DT), gradient boosting (GB), logistic regression (LR), a random forest (RF), and a deep learning transformer-based model called TabTransformer (TabT), and used precision, recall, accuracy, and F1 score to compare the results. The hypertuned TabTransformer model achieved the highest F1 score (discharged: 0.92; deceased: 0.84) out of all the base ML classifiers for both the deceased and discharged datasets. Table 4 shows a comparative analysis of base machine learning models and the TabTransformer model for LoS prediction for the discharged and deceased datasets.

### 3.2. CRFI Results

The CRFI identifies cohorts of risk factors associated with LoS and generates rules based on various patient characteristics. Table 5 illustrates the top sample rules for each category of LoS within the discharged and deceased patient categories. The complete rule set is publicly available in the GitHub repository [20].

#### 3.2.1. CRFI for Discharged Patient Category

In the discharged patient category, for an LoS ≤ 1 week or ≤ 2 weeks, the usage of anticoagulants, antibiotics, and antiviral medications is an important factor, and indicates that timely intervention and appropriate dosages reduce LoS. For an LoS ≤ 3 weeks, some of the most important risk factors observed in the rules were an elevated level of LDH (>225), D-dimer (>500), and CRP (between 6 mg/L and 100 mg/L). These observed rules suggest that abnormal laboratory values prolonged the LoS even with anticoagulant and/or antiviral therapy. For an LoS ≤4 weeks, the most important risk factors observed were a higher lymphocyte count (>1000 cells/µL), an elevated PNN count (1000–7000 mm^3^), comorbidities such as hypertension, and a higher respiratory rate (20–28 bps). The mining results for patients who stayed more than 4 weeks in the hospital show a low platelet count (<50,000), abnormal X-ray, PTT > 14.5, and higher PNN count. We found these patterns along with the usage of antiviral, anticoagulant, and antibiotic medications, which again suggest that abnormal values of the abovementioned risk factors increase LoS even if the medications are provided. The most affected (~40%) age group across all categories was (age ≥56 years and ≤73 years), closely followed by (age ≥38 years and ≤55 years). This observation suggests that age is an important factor in deciding the LoS.

#### 3.2.2. CRFI for Deceased Patient Category

In the deceased patient category, shortness of breath (SOB), a low platelet count (<50000), diastolic blood pressure between (60 and 90), abnormal PTT (>14.5), a higher LDH count (>225), low PaO2 (<80), and elevated TROPONIN between (0 and 0.1) were the most critical factors for the patients who died within 3 weeks of admission to hospital. The most critical risk factors for patients with an LoS ≥ 3 weeks were ALT (0–41), SoB, comorbidities such as diabetes and hypertension, and the Glasgow effect (>14). This pattern suggests that abnormal values of these risk factors not only increase LoS but also increase the severity of COVID-19, leading to eventual patient death. The most affected group was (age ≥56 years and ≤73 years) with 65%, 58.82%, 47.0%, and 56% in rules 1, 2, 3, and 4, respectively. The distribution of younger patients (≤37) across all the rules in the deceased category was very minimal, and an LoS > 3 was most commonly seen for this group.

## 4. Discussion

The current study aimed to analyze cohorts of risk factors obtained from multimodal data using a state-of-the-art deep-learning-based TabTransformer model and association mining. The state-of-the-art TabTransformer model showed excellent results for both deceased and discharged patients in predicting their LoS. The CRFI module was used to analyze a group of risk factors that extend the LoS in hospitals and result in either discharge or death. The CRFI results show the identification of risk factors in cohorts can help in determining LoS and identifying criticalities that influence COVID-19 severity.

Not much work has been carried out on determining LoS for COVID-19 patients using multimodal data. Examples and discussion of the prominent patterns are as follows:Age appears to be a strong risk factor for COVID-19 severity and its outcomes. Statsenko et al. [21] performed a detailed analysis and concluded that elderly patients with COVID-19 are more likely to progress to severe disease. The result of the CRFI for the deceased category identified rules for individuals aged ≥56 years and ≤73 years, while other age category rules were not frequently observed and found to be insignificant. In addition, the mining results for patients who stayed in the hospital for between three and four weeks followed 25% of the rules for patients aged ≥73. These observations validate the fact that age is correlated with COVD-19 severity and a significant factor in deciding LoS.A detailed analysis of CRFI rules for the patients who stayed in the hospital for between 3 and 4 weeks showed that 43% of the rules constituted either hypertension or diabetes; thus, these comorbidities not only increase the LoS in hospitals but also lead to severe COVID-19, leading to increased LoS in the hospital. This was also concluded by Adab et al., 2022 [22].

We also observed many key findings with respect to lab features such as LDH and dimmer and lymphocyte count. A few examples are outlined below:An elevated level of D-dimers is an indicator and major risk factor for thrombosis (blood clotting) and increases the risk of medication and monitoring for a longer time [23]. We observed that for the people who were discharged between 3 and 4 weeks, the CRFI results for D-dimers show that 18% of the rules had a D-dimer value of more than 500 ng/mL FEU, thus increasing the LoS. In addition, in the mining results for patients who stayed more than 3 weeks in the hospital and died, elevated D-dimer values were present in 41% of the rules. This is also validated by the fact that for the people who were discharged within two weeks, the CRFI results show only 4.5% of the rules had a D-dimer value of more than 500 ng/mL FEU, and elevated D-dimer values were not found to be significant according to the CFRI results of patients who stayed less than one week.LDH is another factor that had an elevated level of more than 225 units/L in 23% of the rules, based on the CRFI results of patients discharged from the hospital between 3 and 4 weeks.Wagner et al. [24] concluded that lymphocyte count is one of the most important prognostic factors in determining COVID-19 severity, and our CRFI results for patients who died after spending more than 3 weeks in the hospital found that all the rules with lymphocytes consisted of values between 500 and 1000, while for patients who were discharged within two weeks, 86% of the time, these values were between 1000 and 4000. This again validates the fact that a lower lymphocyte count is critical in determining COVID-19 severity and LoS.During the initial stages of the COVID-19 pandemic, the medical community employed various treatments without substantial evidence to support their efficacy. This was due to the limited understanding of the novel coronavirus and its treatment options at the time. It is important to understand which medications, based on the lessons learned, could be useful to treat infections caused by new viral strains as viable epidemic response strategies. Our study shows that drugs such as Hydroxychloroquine and Favipiravir reduce the patient LoS. The CRFI results for patients who stayed less than a week in the hospital show 51% of the rules consisted of antibiotic medications, while in those discharged in less than 2 weeks, 52% of the rules consisted of antiviral medication. This analysis shows that the usage of antiviral and antibiotic medication effectively reduced patient LoS.

## 5. Limitation and Future Directions

This was a single-institute retrospective cohort study aiming to predict LoS. Using multicenter data, it would be possible to further evaluate the robustness of the proposed framework. Future work will focus on the acquisition of data for different ethnicities and countries. Furthermore, our results do not identify every possible combination responsible for COVID-19 severity and LoS; we only found the prominent ones based on the support and confidence of the association mining model. There is a chance that an important cohort of risk factors was missed because they were not present in the data.

## 6. Conclusions

Predicting a patient’s LoS in a hospital is a complex task due to the multitude of factors that can influence it, including patient history, existing comorbidities, and socio-economic factors. This evaluation demonstrates the efficacy of using a state-of-the-art TabTransformer model in conjunction with association rule mining to predict LoS and assess the impact of different combinations of risk factors on hospital LoS. For COVID-19 patients, LoS can vary greatly depending on factors such as illness severity, comorbidities, and other cohorts of risk factors. The proposed framework can not only be applied for infectious diseases such as COVID-19, but also other critical diseases such as pulmonary and cardiovascular diseases. By accurately predicting LoS, this framework can help hospitals optimize patient care and reduce healthcare costs.

## Figures and Tables

**Figure 1 diagnostics-13-01760-f001:**
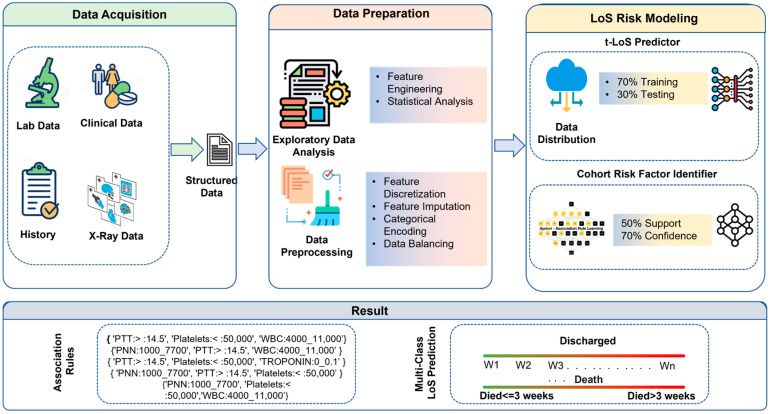
COVID-19 LoS Risk Modeling Workflow.

**Figure 2 diagnostics-13-01760-f002:**
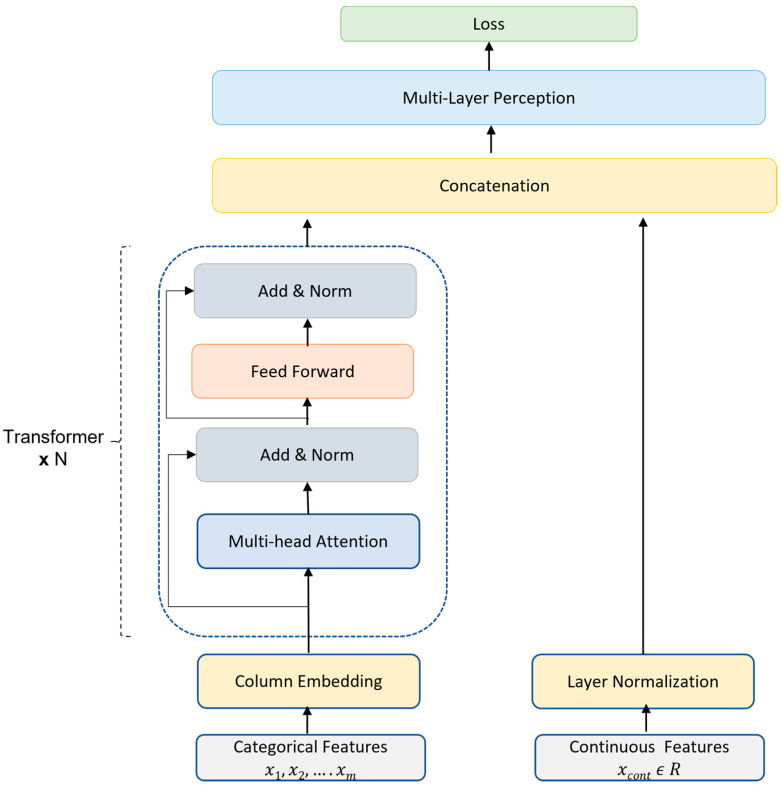
TabTransformer architecture.

**Table 1 diagnostics-13-01760-t001:** Different modalities of data and available features.

Dataset Source	Description	Feature Frequency
General	Contains general information such as demographic data (gender, age, and ethnicity), epidemiological data (date of admission, date of death), and comorbidities such as hypertension, diabetes, COPD, interstitial lung disease, bronchial asthma, liver disease, HIV, cirrhosis, and cardiomyopathies	68
Lab Data	Contains elements related to blood tests such as WBC count, PNN, lymphocyte count, hemoglobin, platelets, creatinine, ALT LDH, FERRITIN, D-DIMER, CRP, PROCALCITONIN, TROPONIN, Pro-BNP, PTT, Vitamin D, and IL6	17
X-ray Data	Contains elements related to X-rays, such as the presence of consolidation and bilateral or unilateral ground-glass opacities	4

**Table 2 diagnostics-13-01760-t002:** Patients’ main characteristics, comorbidities, symptoms, and lab features.

		Patient Characteristics	Details: % of Patients That Qualify
General	Demographic	Gender	Female: 49.7%; Male: 50.3%
		Age	
		Mean	58.8 years
		Median	60 years
		IQR	26.7 years
		Nationality	Egypt: 2%
			Philippines: 1.3%
			Iraq: 0.32%
			Saudi Arabia: 95.7%
			Sudan: 0.36%
			United Kingdom: 0.32%
	Comorbidities	Diabetes	69.2%
		Hypertension	64.3%
		Heart Ischemic	17.2%
		Heart Failure	5.0%
		Cardiomyopathies	1.3%
		COPD	2.0%
		Heart Failure	4.9%
		Interstitial Lung Disease	0.3%
		Bronchial Asthma	15.0%
		Cerebrovascular	4.2%
		Neurologic (Dementia)	4.2%
		Cirrhosis	1.3%
		HIV	0.0%
		Liver Disease	2.0%
		Obesity	5.5%
	Others	Psychiatric History	1.3%
		End Stage Renal	11.0%
		Hemodialysis	4.5%
		Cancer	6.0%
		Solid Organ Transplant	5.5%
		Hematopoietic Cell Transplant	0.0%
		Smoker	0.3%
		Pregnancy	5.0%
		Sick Cell	0.3%
		Shortness of Breath (SOB)	85.7%
		Fever	55.0%
		Hemoptysis	1.0%
		Diarrhea	11.0%
		Cough	72.0%
		Headache	7.5%
		Abdominal Pain	8.0%
		Myalgia	11.0%
		Loss of Smell or Taste	8.0%
		Temperature	100.0%
		Respiratory Rate	13.6%
		Pulse	100.0%
		Nausea or Vomiting	8.0%
		Diastolic BP	100.0%
		Systolic BP	100.0%
		Glasgow	100%
Lab Parameters		LDH	100.0%
		PaCO_2_	100.0%
		HCO_3_	
		PaO_2_	
		pH	
		Lymphocytes	
		PaO_2_	
		WBC	
		ALT	
		PTT	
		D-Dimer	
		Platelets	
		WBC	
		Hemoglobin	
		CRP	
		Ferritin	
		AST	
		NT-proBNP	
		PROCALCITONI	
		TROPONIN	
		Vitamin D	
		IL-6	
		Blood Group	
		INR	
		Fibrinogen	
		PNN	
Medications		Immunomodulators (Tocilizumab)	80.0%
		Antiviral (Favipiravir, Kaletra–Ribavirin–Interferon)	98.0%
		Antibiotic	92.0%
		Anticoagulant (Clexan, Heparine)	87.0%
X-ray		Presence of Consolidation	72.0%
		Presence of Ground-Glass Opacities	
		Bilateral or Unilateral	

**Table 3 diagnostics-13-01760-t003:** Data categorization and original and resampled data.

Classes	LoS in Hospital	Patient Frequency Original	Patient Frequency After SMOTE-N
Deceased	Less than or equal to 3 weeks	36	36
	Greater than 3 weeks	24	36
Discharged	Less than or equal to 1 week	84	84
	1–2 weeks	79	84
	2–3 weeks	37	84
	3–4 weeks	12	84
	Greater than 4 weeks	36	84

**Table 4 diagnostics-13-01760-t004:** Comparative Analysis of the TabTransformer with baseline models for the discharged and deceased datasets.

Classifiers	Discharged Dataset				Deceased Dataset			
	F1	Accuracy	Precision	Recall	F1	Accuracy	Precision	Recall
LR	0.74	0.73	0.77	0.74	0.68	0.68	0.7	0.73
RF	0.73	0.71	0.76	0.72	0.68	0.68	0.7	0.73
DT	0.65	0.65	0.68	0.65	0.62	0.64	0.64	0.66
AB	0.62	0.61	0.63	0.62	0.61	0.64	0.61	0.62
GB	0.54	0.52	0.61	0.53	0.50	0.5	0.6	0.6
TabT *	0.92	0.73	0.83	0.93	0.84	0.77	0.75	0.98

LR (logistic regression), RF (random forest), DT (decision tree), AB (AdaBoost), GB (gradient boost), TabT (TabTransformer), * best-performing model.

**Table 5 diagnostics-13-01760-t005:** Top sample rules for discharged and deceased categories of LoS.

Dataset Type	LoS Category	Association Rules
Discharged dataset	LoS ≤ 1 Week	{Anticoagulant, Cough, Antibiotics, Antiviral}
		{Cough, LDH > 225, Antibiotics, Antiviral}
		{Anticoagulant, SOB, Immunomodulators, LDH > 225, Antibiotics, Platelets < 50,000}
		{PaO2 (0 to 80), Anticoagulant, SOB, LDH > 225, Antibiotics}
	LoS 1–2 Weeks	{Fever, DIMER (0 to 500), Immunomodulators, Antibiotics, Temperature (36 to 37.6)}
		{PaO2(0 to 80), Fever, Immunomodulators, LDH > 225, Antibiotics, Antiviral}
		{Anticoagulant, Fever, FERRITIN < 792, Immunomodulators, Glasgow > 14, Platelets < 50,000}
		{Anticoagulant, Fever, SOB, HTN, Glasgow > 14, Antiviral}
	LoS 2–3 Weeks	{Fever, DIMER > 500, LDH > 225, Antiviral}
		{Anticoagulant, Fever, HTN, Diastolic BP (60 to 90), Antiviral}
		{Anticoagulant, Fever, HTN, Immunomodulators, Diastolic BP (60 to 90)}
		{CRP (6 to 100), Fever, LDH > 225, Antiviral}
	LoS 3–4 Weeks	{Anticoagulant, Lymphocytes (1000 to 4000), Antibiotics, Respiratory Rate (20 to 28), PNN (1000 to 7700)}
		{Anticoagulant, HTN, Immunomodulators, Lymphocytes (1000 to 4000), Antibiotics, Respiratory Rate (20 to 28), Antiviral}
		{HTN, Immunomodulators, Lymphocytes (1000 to 4000), Antibiotics, Respiratory Rate (20 to 28), Antiviral}
		{Anticoagulant, Immunomodulators, Lymphocytes (1000 to 4000), PNN (1000 to 7700)}
	LoS ≥ 4 Weeks	{Immunomodulators, Platelets < 50,000, Antiviral, abnormal X-ray}
		{Antibiotics, PTT > 14.5, Platelets < 50,000}
		{Anticoagulant, Immunomodulators, Antiviral}
		{PNN (1000 to 7700), PTT > 14.5, Platelets < 50,000, Antiviral}
Deceased dataset	LoS ≤ 3 Weeks	{SOB, Antibiotics, PTT > 14.5, Platelets < 50,000, TROPONIN (0 to 0.1)}
		{SOB, Antibiotics, Glasgow > 14, PNN: 1000_7700, Antiviral}
		{LDH > 225, Diastolic BP (60 to 90), Glasgow > 14, Antiviral}
		{PaO2 (0 to 80), Cough, Antibiotics, Glasgow > 14, Platelets < 50,000}
	LoS > 3 Weeks	{ALT (0 to 41), Diabetes, HTN, Immunomodulators, Antibiotics}
		{Ph (7.35 to 7.45), ALT (0 to 41), HTN, Immunomodulators, Antibiotics, Platelets < 50,000}
		{ALT (0 to 41), SOB, HTN, Immunomodulators, Antibiotics, PTT > 14.5}
		{ALT (0 to 41), SOB, HTN, Immunomodulators, Glasgow > 14, Antiviral}

## Data Availability

The datasets, libraries, and any supporting tools used or analyzed during the current work are accessible upon reasonable request from the corresponding author.

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
