# Peer review of "Towards Predicting Length of Stay and Identification of Cohort Risk Factors Using Self-Attention-Based Transformers and Association Mining: COVID-19 as a Phenotype"

_diagnostics, 2023, doi:10.3390/diagnostics13101760_

Round 1

Reviewer 1 Report

Please find the Comments in the attached PDF-file

Please find the Comments in the attached PDF-file

Reviewer 2 Report

In this manuscript the Authors aimed to forecast patients’ Predicting Length of Stay (LoS) using a deep learning model and analyze cohorts of risk factors minimizing or maximizing LoS.

The topic investigated is of significant interest for human health studies.

A good amount of evaluations have been done .

The description of methodology results accurate and precise.

The findings have been properly reported and the obtained data well discussed by using available literature and the recent findings by other authors.

The Conclusion section provides a clear overview of the findings and their usefulness, evel if it could be further improved.

As specific comments, in order to further improve the quality of the paper, I suggest to:

- Try to add into the Abstract section futher details.

- An overall check of the English language is suggested in order to further improve the paper's quality.

-Some additional recently published references may add value to the Introduction section;

- Check all acronyms used, spell at first use;

- Conclusion section could be improved.

- The references have been reported in an appropriate form and edited according to the journal’s guidelines, but some references need revision.

So, based on my opinion, I think that this paper need minor revisions.

Minor editing of English language required

Reviewer 3 Report

This study design in this research focuses on a detailed analysis of interaction and association between different risk factors, and develops a framework for predicting the  Predicting Length of Stay (LoS) of COVID-19 patients using a state-of-the-art transformer-based architecture and  identifies groups of influencing factors occurring together that affect LoS using pattern recognition techniques by mining multi-modal patient data such as lab data, x-ray data of  Covid-19 patients. 

The main contribution of this paper can be summarized as follows: 

- A state of art attention-based tab transformer model to predict patient LoS using multiple modalities such as clinical features, patient demographic data, and X-ray reports. 

The authors present a framework where the result of machine learning based methods for  LoS prediction can be analyzed with association mining rules to identify the cohort of risk factors affecting the LoS in hospitals.

The authors performed experiments with five ML models: AdaBoost (AB), Decision Tree  (DT), Gradient Boosting (GB), Logistic Regression (LR), Random Forest (RF), and a deep  learning transformer-based model called Tab Transformer (TabT), and used precision, recall, accuracy, and F1 score to compare the results.

- Age appears to be a strong risk factor for COVID-19 severity and its outcomes

- A detailed analysis on CRFI rules on the patients who stayed in hospital between  3 and 4 weeks showed that 43 % of the rules constitute either hypertension or diabetes,

- The elevated level of D-dimers is an indicator and major risk factor for thrombosis  (blood clotting) and increases the risk of medication and monitoring for a longer time

- LDH is another factor that has an elevated level of more than 225 units/L in 23%  of the rules, based on the CRFI results of patients discharged from the hospital between 3 and 4 weeks.

The work is well done . I have some remarks:

- I suggest a English language editing

- The text format have be revised

- I suggest a English language editing
